# Sleep Spindle Characteristics and Relationship with Memory Ability in Patients with Obstructive Sleep Apnea-Hypopnea Syndrome

**DOI:** 10.3390/jcm12020634

**Published:** 2023-01-12

**Authors:** Qilin Zhu, Fei Han, Jin Wang, Chaohong Chen, Tong Su, Qiaojun Wang, Rui Chen

**Affiliations:** 1Department of Respiratory and Critical Care Medicine, The Second Affiliated Hospital of Soochow University, Suzhou 215004, China; 2Department of Respiratory Medicine, The Third Affiliated Hospital of Nantong University, Nantong 226001, China

**Keywords:** obstructive sleep apnea syndrome, EEG, power spectral density, omega complexity, memory ability

## Abstract

Obstructive sleep apnea syndrome (OSAS) causes intermittent hypoxia and sleep disruption in the brain, resulting in cognitive dysfunction, but its pathogenesis is unclear. The sleep spindle wave is a transient neural event involved in sleep memory consolidation and synaptic plasticity. This study aimed to investigate the characteristics of sleep spindle activity and its relationship with memory ability in patients with OSAS. A total of 119 patients, who were divided into the OSAS group (*n* = 59, AHI ≥ 15) and control group (*n* = 60, AHI < 15) according to the Apnea Hypopnea Index (AHI), were enrolled and underwent polysomnography. Power spectral density (PSD) and omega complexity were used to analyze the characteristics of single and different brain regions of sleep spindles. Memory-related cognitive functions were assessed in all subjects, including logical memory, digit ordering, pattern recognition, spatial recognition and spatial working memory. The spindle PSD of the OSAS group was significantly slower than the control group, regardless of the slow, fast, or total spindle. The complexity of the spindles in the prefrontal and central region decreased significantly, whereas it increased in the occipital region. Sleep spindle PSD was positively correlated with logical memory and working memory. Spindle complexity was positively correlated with immediate logical and visual memory in the prefrontal region and positively correlated with immediate/delayed logical and working memory in the central region. In contrast, spindle complexity in the occipital region negatively correlated with delayed logical memory. Spindle hyperconnectivity in the prefrontal and central regions underlies declines in logical, visual and working memory and weak connections in the occipital spindles underlie the decline in delayed logical memory.

## 1. Introduction

Obstructive sleep apnea hypoventilation syndrome (OSAS) is a serious sleep-disordered breathing disorder in which patients experience recurrent apnea and hypoventilation during sleep [1]. Due to the repeated arousal and arousal responses of the cerebral cortex at night, the normal sleep structure and rhythm are disrupted, and the patients have sleepiness during the day and even cognitive decline [2].

Memory function is closely related to brain activity during sleep [3]. Obstructive sleep apnea hypoventilation syndrome affects memory ability, which can be found in patients’ self-reports, such as difficulty concentrating and forgetfulness. A study of middle-aged OSAS patients showed significant memory decline [4]. Sleep fragmentation caused by OSAS affects sleep-dependent memory consolidation [5]. Studies of electrical activity in the brain have found that non-rapid eye movement sleep (NREM) contributes to procedural memory consolidation [6]. From the perspective of microscopic sleep structure, sleep spindles in NREM sleep are related to sleep stability and memory ability.

The physiological functions of the sleep spindle mainly maintain sleep stability [7] and participate in the integration of memory [8]. Normal sleep stages include non-rapid-eye-movement sleep (NREM) and rapid-eye-movement (REM) sleep. Non-rapid-eye-movement sleep is divided into N1, N2 and N3. Sleep spindles are transient EEG rhythms produced by the reticular nucleus of the thalamus during non-REM sleep, consisting of 11–16 Hz fluctuating oscillations that last 0.5–3 s and are particularly prevalent during N2 [9]. Previous studies reported decreased overall spindle density (11–16 Hz), frontal slow spindle density (11–13 Hz), centroparietal fast spindle density (13–16 Hz) and overall spindle frequency in patients with OSAS compared to controls [10,11,12]. Numerous studies on the correlation between sleep disorders and sleep spindles have shown that patients with OSAS have general sleep structure disruption and sleep spindle reduction [13,14,15]. It has been found that abnormal sleep spindle activity is caused by the dysfunctional synchronization of EEG activity across brain regions [16,17]; promoting sleep spindles can improve memory function [18].

The widespread dysfunctional synchronization of neural oscillations may be one of the key factors in disease progression, requiring many brain regions to work together to perform even the simplest tasks [19]. The connectivity between brain regions at the macroscopic scale can be quantitatively assessed by modern brain imaging techniques. Brain activity can be studied by the electroencephalogram (EEG) with high temporal resolution. The spatial complexity of brain signals refers to the degree of heterogeneity in the signals of different “nodes” of the brain, which is inversely proportional to the overall level of functional connectivity between these nodes [20]. Ref. [21] proposed omega complexity to quantify the concept of spatial complexity. Compared with the traditional functional connectivity analysis, the spatial complexity analysis technique of omega complexity is more convenient and less computational. However, previous studies on patients with OSAS mainly focus on the changes in oscillatory neural activity in a single brain region or a single frequency band without considering the synchronization between different brain regions [13,22]. Previous studies have also investigated the relationship between spindle parameters and memory ability [23]. These studies focused on time domain characteristics such as number, density and frequency, which could not fully reflect the physiological characteristics of the sleep spindle.

Therefore, this study first aims to reveal the oscillatory characteristics of sleep spindle activity and the connectivity characteristics between different brain regions in OSAS patients. The second aim is to explore the relationship between sleep spindle characteristics and memory function, providing evidence for clinical research.

## 2. Methods

### 2.1. Participants

The study was prospectively performed at the Second Affiliated Hospital of Soochow University from January 2021 to December 2021. A total of 145 patients who received polysomnography (PSG) monitoring due to snoring were enrolled. According to the Guidelines for the Diagnosis and Management of Obstructive Sleep Apnea Hypoventilation Syndrome (2011 Revised Edition), the apnea-hypopnea index (AHI) was used as a diagnostic index [24]. Two cases of central sleep apnea, nine patients due to recent oral sedative drugs, and seven patients due to unwillingness to perform scale testing were excluded. As the sample size of females was limited (eight excluded), this study was conducted on males. Finally, a total of 119 people were included in the study. Patients with AHI ≥15 events/h were placed into OSAS group (*n* = 60); based on previous studies [25,26], patients with AHI < 15 events/h were placed into control group (*n* = 59).

Inclusion criteria: (1) age from 18 to 60; (2) junior high school or above, right-handed; (3) complete medical history and PSG parameters; (4) did not receive OSAS surgery or ventilator treatment. Exclusion criteria: (1) central sleep apnea, insomnia and other sleep-related diseases; (2) any neurological or psychological disorders that interfere with cognitive function tests, such as Parkinson’s disease, Alzheimer’s disease, anxiety, depression, or taking psychiatric medications; (3) patients with color blindness and color weakness, unable to complete computer software evaluation; (4) suffering from diseases that seriously affect quality of life, such as severe heart and lung diseases, cancer, stroke and severe liver and kidney insufficiency.

The study was conducted in accordance with the Declaration of Helsinki principles, and all procedures were carried out with adequate understanding and written informed participant consent. Research ethics approval was obtained from the Ethics Committee of the Second Hospital of Soochow University (JD-LK-2018-004-02). All participants gave written informed consent.

### 2.2. Measurement

#### 2.2.1. Basic Information

Detailed basic information and medical history of all patients were collected. It mainly included age, gender, education level, body mass index (BMI), drinking history, smoking history, hypertension, diabetes history and whether there was apnea at night, waking up at night, sleepiness, memory loss, morning fatigue, morning headache, morning dry mouth, insomnia.

#### 2.2.2. Epworth Sleepiness Scale

The ESS is a useful tool for assessing sleepiness, which is often used to evaluate the severity of daytime sleepiness in patients with sleep-disordered breathing [27]. The total score ranges from 0 to 24. A score of 10 indicates drowsiness. The ESS score of the patients was performed by the specialists of the sleep center.

#### 2.2.3. Memory Ability Assessment

The Cambridge neuropsychological test automated battery (CANTAB) is a reproducible and validated computerized cognitive testing system [28]. All subjects were assessed by the same trained doctor using the same guide language from 18:00 to 19:00 on the day of sleep monitoring. This study primarily used tests related to memory consolidation.

Digit Ordering Test (DOT)

The DOT is used to assess working memory, consisting of 6 entries of progressively increasing length (3–8 digits). Each entry has two strings of numbers, the second of which has duplicate numbers. After the tester reads the entries at a rate of 1 digit per second, the subject is asked to immediately repeat the digits in order from smallest to largest, and if both test questions of the same length are incorrect, the test is terminated.

Logical Memory Test (LMT)

Logical memory tests assess logical memory, including immediate and delayed logical memory. Participants were asked to retell the story immediately after reading a short paragraph and trying to remember it. Participants were asked to retell the story again after 30 min, which was the LMT delay score.

Pattern Recognition Memory (PRM)

Pattern Recognition Memory assesses visual memory. The subjects were asked to memorize as many pictures as possible, and then two groups of 12 similar pictures appeared on the screen, and they were asked to quickly select the pictures that had appeared to assess immediate visual memory.

Spatial Recognition Memory (SRM)

Spatial Recognition Memory is used to assess subjects’ spatial recognition memory, focusing on the memory of spatial position changes. Five empty squares of different spatial locations and the same size appear on the screen one by one, and the participant is asked to remember the spatial locations where all the squares have appeared. The test was conducted in 2 sets of 5 squares each. The correct rate of each group was recorded, and the correct average rate was calculated.

Spatial Working Memory (SRM)

Spatial Working Memory is used to test spatial working memory. Subjects were asked to click on colored squares to find the hidden blue box used to fill the empty column on the right side. It is worth noting that each square does not appear twice in a row, and it is an error to repeatedly click on a square where a blue box has appeared.

### 2.3. Sleep EEG

Polysomnography was examined after inquiring about general clinical data and medical history. All patients’ overnight polysomnography was completed in the sleep center (Alice 6, Philips, New York, NY, USA). The recording started at 10:00 pm and ended at 6:00 am the next day, and was monitored continuously for more than 7 h. Two full-time technicians manually assessed the data quality and labeled the sleep stage according to American Academy of Sleep Medicine standards. Since sleep spindles are characteristic waves in the N2 stage, the EEG data in the N2 stage were analyzed.

The PSG recordings included eight electrode (F3, F4, C3, C4, O1, O2, A1 and A2) electroencephalograms (EEG) placed according to the international 10–20 system. During EEG recording, all electrodes were referred to a reference electrode placed in the left mastoid process. During off-line analysis, the effective electrode placed in the right mastoid process was again referred to; that is, the signal recorded by the reference electrode was subtracted from each lead signal and converted to the average value of the bilateral mastoid process as a reference. The filter band pass was 0.05–100 Hz, the sampling frequency was 200 Hz and the impedance between all electrodes and scalp was less than 10k ω.

### 2.4. The Power Spectrum Density Analysis

The sleep spindle oscillation characteristics can be reflected by analyzing the energy composition of the specific frequency bands. MATLAB software (Version R2013b, MathWorks, Natick, MA, USA) was used for power spectral densities. In this study, the power spectral densities (PSD) of sleep spindles (11–16 Hz), fast spindles (13–15 Hz) and slow spindles (11–13 Hz) were calculated. The Welch method was used to calculate the PSD of the spindle waves of each channel EEG signal (F3, F4, C3, C4, O1, O2), which is widely used in calculating the power spectra of EEG [29]. In the calculation process, the parameters of the Welch method are set as follows: the length of Hamming window is 256 data points and the length of the overlapping window is 128 data points.

At length *L* signal *x_L_*[*n*] (*n* = 0, 1, 2, …, *L* − 1), if the sampling frequency is *fs*, the formula for calculating the power spectrum estimation of the corresponding *N* point is as follows:(1)$${\hat{P}}_{xx}\left(f_{k}\right)=\frac{{\left|X_{L}\left(f_{k}\right)\right|}^{2}}{f_{s}L}.$$
where
(2)$$X_{L}\left(f_{k}\right)=\sum_{n=0}^{L-1}x_{L}\left[n\right]e^{-2\pi jkn/N},$$
(3)$$f_{k}=\frac{kf_{s}}{N},k=0,1,\ldots ,N-1.$$

### 2.5. Omega Complexity Analysis

The EEG data from each subject were transformed to the frequency domain using Fourier Transform. The omega complexity of spindles was calculated as the average value within three frequency bands: slow spindle (11–13 Hz); fast spindle (13–16 Hz); total spindle (11–16 Hz). Regional complexity was obtained by calculating a cross-spectral matrix using electrodes in the prefrontal (F3 and F4), central (C3 and C4) and occipital (O1 and O2) regions. Principal component analysis of these cross-spectral matrices yielded a spectrum of eigenvalues. These were normalized to assess the relative contribution of each principal component to the total variance. The calculation process of omega was as follows. First, the cross-spectral matrix *C* of global complexity and local complexity was calculated, respectively:

(4)$$C=\frac{1}{N}\sum_{i=1}^{N}u_{i}*u_{i}^{T}$$
where *K* is the number of electrodes, *N* is the EEG signal length.

Then, the eigenvalue *λ*_1_, …, *λ_k_* of matrix *C* was calculated. Next, the normalized feature $\lambda_{i}^{\prime }$ was calculated.
(5)$$\lambda_{i}^{\prime }=\frac{\lambda_{i}}{\sum \lambda_{i}}$$

According to the definition of omega:(6)$$\log\left(\Omega \right)=-\sum \lambda_{i}^{\prime *}\log\left(\lambda_{i}^{\prime }\right)$$

Therefore, the omega spatial complexity can be calculated:



(7)
$$\;\mathrm{Omega}\;=\exp\left\{-\sum_{i=1}^{K}\lambda_{i}^{+}\log\left(\lambda_{i}^{\prime }\right)\right\}$$



The omega complexity can be viewed as a measure of the spatial complexity of a given set of EEGs, which ranges in value from 1 to *K*. A smaller omega value indicates that the number of modes existing between the computed signals is small with a single-mode and a high degree of synchronization; a larger omega value indicates that the number of modes existing between the computed signals is large and the degree of synchronization is poor. For example, if the value of omega tends to *K*, it is known from the knowledge of information theory that the calculated signal has a uniform distribution of modes with the highest number of modes, the highest complexity and the worst synchronization; on the contrary, if the value of omega tends towards 1, it indicates that the distribution of the signal has only one mode and the highest synchronization is achieved. The omega complexity values for each frequency point were averaged over the epochs [30].

### 2.6. Statistical Analysis

Statistical analysis was performed using SPSS (23.0; SPSS, Inc., Chicago, IL, USA). The goodness-of-fit test was used for non-normally distributed data. Two-way analyses of mixed repeated-measures ANOVA with a between-subject factor for the group (OSAS and control groups) and a within-subject factor for frequency band (slow, fast and total spindles) were performed for power spectral density of global. The power density of each electrode was tested using an independent sample *t*-test. A 2 (OSAS and control groups) × 3 (slow, fast and total spindles) × 3 (prefrontal, central and occipital) mixed repeated-measures ANOVA was performed for the omega complexity. A Greenhouse–Geisser correction of the ANOVA assumption of sphericity was applied where appropriate. The Bonferroni correction method was used to correct multiple comparisons. Spearman correlation was used to analyze the correlation between sleep spindle features (energy and omega complexity) and behavioral data on memory tasks. The level of significance was set at *p* < 0.05.

## 3. Results

### 3.1. Demographics

The demographic and clinical characteristics are shown in Table 1. There were no significant differences in age, drinking history, smoking history, the history of previous diseases, nocturnal wakeup, nocturnal urination, leg movement, dream, nightmare, morning fatigue, morning headache and early wakeup between the OSAS and control groups, all *p* > 0.05. However, the ESS scores were significantly higher for the OSAS group than the control group, *p* < 0.05. The symptoms of apnea, dry mouth, drowsiness and memory loss in the OSAS group were more obvious than those in the control group, all *p* < 0.05.

The results of the memory function between OSAS Group and control are shown in Table 2.

### 3.2. The Sleep Spindle PSD Results

The PSD of sleep spindles of the whole brain was first analyzed (Figure 1). 

The results of the mixed repeated measures ANOVA showed a significant frequency band main effect, (F (2, 174) = 42.91, *p* < 0.000, $\eta_{p}^{2}$ = 0.33). Post hoc analysis revealed that the PSD of the slow spindle (*M* = 0.90 µV, 95% CL (0.81, 0.99)) was significantly higher for the fast (*M* = 0.72 µV, 95% CL (0.65, 0.79), *p* = 0.000) and total spindles (*M* = 0.81 µV, 95% CL (0.74, 0.89), *p* = 0.001). The total spindle was significantly higher for the fast spindles.

The results also showed a significant group main effect (F (2, 174) = 14.77, *p* = 0.000, $\eta_{p}^{2}$ = 0.15). Post hoc analysis revealed that the PSD was significantly higher in the control (*M* = 0.96 µV, 95% CL (0.84, 1.07)) than in the OSAS (*M* = 0.66 µV, 95% CL (0.56, 0.76), *p* = 0.000).

A significant interaction effect between two group (OSAS/control) × three frequency bands (slow, fast, total spindles) was found, F (2, 174) = 6.02, *p* = 0.003, $\eta_{p}^{2}$ = 0.07. Simple effects analysis of the frequency band revealed significant differences, *p* < 0.05. Further analysis of fast spindles and slow spindles revealed the PSD of slow spindles was significantly higher for the control group than the OSAS group ((1.07 ± 0.36) vs. (0.73 ± 0.45), *t* (87) = 3.95, *p* < 0.000); the PSD of fast spindles was also significantly higher for the control group than the OSAS group ((0.82 ± 0.25) vs. (0.61 ± 0.39), *t* (87) = 3.03, *p* = 0.003). Simple effects analysis of the groups revealed there were significant differences in the control (F (2, 78) = 18.43, *p* = 0.000, $\eta_{p}^{2}$ = 0.32) and OSAS groups (F (2, 96) = 47.63, *p* = 0.000, $\eta_{p}^{2}$ = 0.50). Post hoc analysis revealed that in the control group there was a significant difference between slow (1.07 ± 0.06) and fast (0.82 ± 0.4) and total (0.97 ± 0.05) spindles. However, there was no significant difference between fast wave and total spindles. Post hoc analysis revealed that in the OSAS group there was a two-by-two difference between all three frequency bands, slow (0.73 ± 0.06) > total (0.66 ± 0.06) > fast (0.61 ± 0.05) spindles, all *p* = 0.000.

Then, the PSD of the spindle of the single electrode for the control and OSAS groups was analyzed, as shown in Table 3. The PSD of the slow spindle (11–13 Hz), fast spindle (13–16 Hz) and total spindle (13–16 Hz) in F3, F4, C3, C4, O1 and O2 electrodes was significantly slower for the OSAS group than that in the control group, all *p* < 0.05.

### 3.3. Omega Complexity Results

A two group (OSAS/control) × three brain regions (prefrontal, central, occipital) × three frequency bands (slow, fast, total spindle) mixed analysis of ANOVA was performed for omega complexity. The results of omega complexity are shown in Figure 2.

The main effect of brain regions was significant, F (1, 87) = 66.62, *p* < 0.000, $\eta_{2}^{p}$ = 0.043. Post-hoc analysis revealed that the complexity of the prefrontal region (1.63 ± 0.004) was higher than the central (1.64 ± 0.005) and occipital (1.64 ± 0.005) regions, with *p* < 0.05. However, there was no significant difference between the central and occipital regions, *p* > 0.05. The main effect of frequency bands was also significant, F (1, 87) = 81.15, *p* < 000, $\eta_{2}^{p}$ = 0.48. Post-hoc analysis revealed that the complexity of the slow spindle (1.76 ± 0.006) was higher than the fast (1.60 ± 0.01) and total (1.56 ± 0.01) spindles, with *p* < 0.05. However, there was no significant difference in complexity between the fast and total spindles, *p* > 0.05.

The analysis of results revealed significant interaction effects between the group and brain regions, F (2, 174) = 14.15, *p* < 0.001, $\eta_{2}^{p}$ = 0.14. Simple effect analysis indicated a significant difference in the central and occipital regions. Further analysis revealed that the complexity of the control group (1.66 ± 0.06) was higher than that of OSAS (1.53 ± 0.13) in the central region, *p* < 0.001, whereas the complexity of control (1.49 ± 0.08) was slower than that of OSAS (1.63 ± 0.15) in the occipital region, *p* < 0.001.

It is worth noting that the analysis of results showed that the triple interaction among the group, brain regions and frequency bands was extremely significant, F (4, 348) = 4.38, *p* = 0.002, $\eta_{2}^{p}$ = 0.05. The results showed significant differences between the two groups in the distribution of some brain regions in the slow, fast and total spindles. Simple effects analysis found significant differences in the brain’s three frequency bands: prefrontal, central and occipital regions. Simple effect analysis showed that the complexity of the control group was higher than that of the OSAS group in the central region of the spindle. However, the complexity in the occipital region was slower than that in the OSAS group. In the fast spindle, the control group had higher complexity in the frontal and central regions than the OSAS group. However, the complexity in the occipital region was slower than that in the OSAS group. In the total spindle, the control group had higher complexity in the frontal and central regions than the OSAS group. However, the complexity in the occipital region was slower than that in the OSAS group.

In conclusion, a pairwise separation of omega complexity was found in the central and occipital regions of the brain, with an increase in the complexity of fast spindles and a decrease in the complexity of fast and total spindles in the control group.

### 3.4. Correlation of Sleep Spindle Features with Memory Ability Results

Firstly, the correlation between memory ability and the PSD of sleep spindle activity was analyzed. The results showed a significant positive correlation between spindle PSD and logical memory, whether fast (*r* = 0.40, *p* = 0.004), slow (*r* = 0.38, *p* = 0.006) or total sleep (*r* = 0.40, *p* = 0.005) spindles (Figure 3A). The results also showed that working memory was positively correlated with fast (*r* = 0.29, *p* = 0.048) and total (*r* = 0.28, *p* = 0.047) spindles (Figure 3B).

Secondly, the correlation between memory ability and the omega complexity of sleep spindle connection in different brain regions (prefrontal, central, occipital regions) was analyzed. In the prefrontal region (F3-F4), spindle complexity was positively correlated with visual memory regardless of whether it was slow (*r* = 31, *p* = 0.029), fast (*r* = 0.42, *p* = 0.003) or total (*r* = 0.37, *p* = 0.01) spindles (Figure 4A); the immediate logical memory is positively correlated with fast spindles (*r* = 0.29, *p* = 0.04) (Figure 4B). In the central region (C3-C4, Figure 4E), slow (*r* = 0.29/0.35, *p* = 0.04/0.01), fast (*r* = 0.29/0.30, *p* = 0.04/0.03) or total (*r* = 0.29/0.34, *p* = 0.04/0.012) spindle complexity was positively correlated with immediate and delayed logical memory (Figure 4D). Slow (*r* = 0.31, *p* = 0.03) and total (*r* = 0.28, *p* = 0.048) spindle complexity in the central region were also positively correlated with working memory (Figure 4F). In the occipital region (O1-O2), there was a negative correlation with delayed logical memory no matter the slow (*r* = −0.44, *p* = 0.002), fast (*r* = −0.35, *p* = 0.01) or total (*r* = −0.38, *p* = 0.007) sleep spindle complexity (Figure 4C).

## 4. Discussion

Previous studies have suggested that sleep spindles are correlated with memory function, which may be involved in the occurrence and development of memory function impairment in OSA patients [31,32,33]. To investigate the characteristics of the sleep spindle and its relationship with memory function in patients with OSAS, the power spectral density and brain connectivity of the spindle was quantified using the Fourier transform and omega complexity methods.

The PSD results showed that the PSD values of the sleep spindle were significantly slower in the OSAS group compared to the control group. After the sleep spindles were divided into fast and slow spindles, the PSD values of the fast and slow spindles remained significantly different between the two groups, with the OSAS group slower than the control group. In addition, the topographic differences of the sleep spindles between the two groups were evaluated, and the OSAS group significantly reduced at each electrode (F3, F4, C3, C4, O1, O2), *p* < 0.05. The results of the present study on fast and slow spindles were partially similar to those of previous studies, in which patients with respiratory disorders had significantly slower and fast spindle density compared with controls [14,34]. Interestingly, we found that there was no significant difference between fast and total sleep spindles in the control group, while the power of the fast spindle was higher than that of the total spindle in the OSAS group, which provides evidence that the fast spindle is more sensitive to patients with OSAS pathology than the slow spindle [35].

The brain is complex at multiple levels of temporal and spatial scale, consisting of interconnected feedback loops [36]. Studies of brain signal complexity may provide insight into the relationship between physiological complexity and cognitive performance. Compared with traditional EEG spectral power analysis methods, omega complexity to quantify the spatial complexity of EEG data can be used to assess the number of independent electrophysiological sources and the degree of global synchronization between spatially distributed brain regions [37]. The results showed the omega complexity of sleep spindles in frontal (F3-F4), central (C3-C4) and occipital (O1-O2) regions separately, aiming to discover more in-depth features in patients with OSAS. Omega complexity describes the synchrony of sleep spindles in different brain regions during brain activity, which can explain the connectivity or coordination problems of spindle information transmission in OSAS patients during sleep. We found that the complexity between the left and right brain regions in the central region (C3-C4) was slower in the OSAS group than in the control group in both fast and slow sleep spindles. In the prefrontal region (F3-F4), the spindle complexity of the OSAS group was also slower than that of the OSAS group, but only in fast and total spindles, where the significance of the fast spindle was higher than that of the total spindle. The decrease in omega complexity values indicates hyper-connectivity between the central and frontal spindles in OSAS patients. However, the opposite result was found in the occipital region (O1-O2). The omega complexity of the OSAS group was higher than that of the control group in slow, fast and total frequency ranges. Increased complexity means less collaboration between the independents of simultaneous processes [38]. Higher values of omega complexity indicate a reduced degree of synchronization between spatially distributed cortical regions. The high omega complexity in the occipital region of the OSAS group compared to the control group for both fast and slow fusiform waves indicates that these patients have more independent and parallel functional processes, suggesting that respiratory impairment may lead to loosening coordination, cooperation and connectivity between occipital cortical regions [39]. There is a barrier to left-right brain information transfer between O1-O2, and connectivity is weak in the left and right brain regions.

Both slow and fast sleep spindles are characteristic of NREM sleep, and both oscillations are markers of neuronal plasticity with roles in memory function [40]. The correlation between the sleep spindle and memory function showed that the slow, fast and total spindle PSD of the whole brain was positively correlated with delayed memory. Further analysis showed omega complexity in the central region was significantly negatively correlated with memory delay, whereas it was positively correlated with memory delay in the occipital area. The correlation results also showed that the prefrontal (F1-F2) spindles’ PSD was positively correlated with verbal working memory, while the sleep spindles were still significantly correlated after being classified into fast and slow spindles. The prefrontal (F1-F2) spindle complexity correlation results found that there was a negative correlation with verbal working memory, suggesting that hyper-connectivity may be associated with decreased verbal working memory. In addition, we found that the increase in spindle complexity in the occipital region was positively correlated with visual memory as well as being present in both fast and slow fusiform waves.

Visual memory refers to the input, encoding, storage and extraction of information from the visual channel; that is, the individual’s ability to recognize, retain and reproduce visual experiences. The increase in occipital complexity indicates an obstacle to left-right brain information transmission that may lead to decreased visual memory. The fast spindle in particular was found to activate brain networks involved in consolidating sleep-related memories [41]. Logical memory is a complex form of memory in which information is understood, analyzed and judged and situational memory is the memory of everyday events and experiences at a specific time and place [42]. Previous research by our team found that hypercapnia had negative impacts on the logical memory and working memory of OSAS patients, especially on delayed logical memory, verbal working memory and spatial working memory impairment [43]. Stevens et al. found that spindle activity was diminished in the frontal and central regions during NREM, causing impaired learning consolidation [44]. The present study further finds that the hyper-connectivity of the sleep spindle between the left and right brain areas in the central and prefrontal regions is related to visual memory and logical memory.

There are still some limitations in this study. Firstly, this is a cross-sectional observational study, and longitudinal correlation between cognitive function and the sleep spindle in OSAS patients could not be observed, which will be reported in follow-up. Secondly, the sample age of this study was between 23 and 60 years old; few patients who were over 65 years old came to the clinic. The cognitive function impairment of OSAS may be more severe in the elderly, and the sample size and age range will be expanded later for further study. In addition, it has been reported that gender has a significant effect on the power spectral density [45]. The subjects of this study were all males, and future research gaps in females should be added.

## 5. Conclusions

The PSD of sleep spindles was reduced in OSAS patients. Further analysis revealed that sleep spindle connectivity was enhanced in the occipital region and reduced in the prefrontal region, which may be associated with logical memory and working memory impairment in OSAS patients.

## Figures and Tables

**Figure 1 jcm-12-00634-f001:**
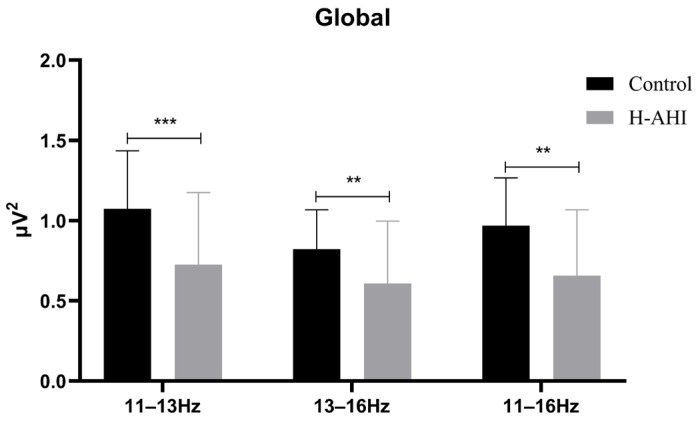
Power spectral density results. Note: **, *p* < 0.01; ***, *p* < 0.001.

**Figure 2 jcm-12-00634-f002:**
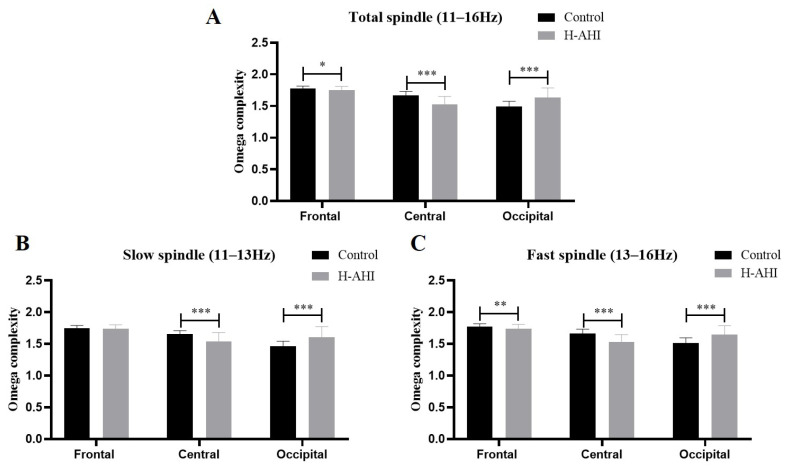
Omega complexity of sleep spindle results. (**A**) Total spindle; (**B**) Slow spindle; (**C**) Fast spindle. Note: *, *p* < 0.05; **, *p* < 0.01; ***, *p* < 0.001.

**Figure 3 jcm-12-00634-f003:**
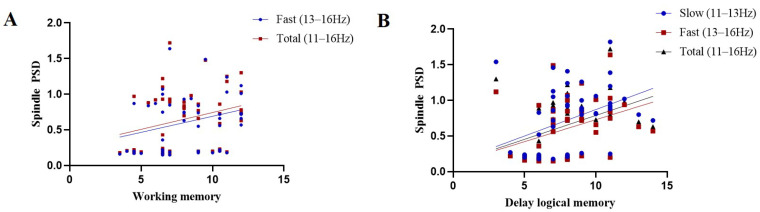
Correlation results of sleep spindle power spectral density and memory function. (**A**) the relationship sleep spindle power with logical memory; (**B**) the relationship sleep spindle power with working memory.

**Figure 4 jcm-12-00634-f004:**
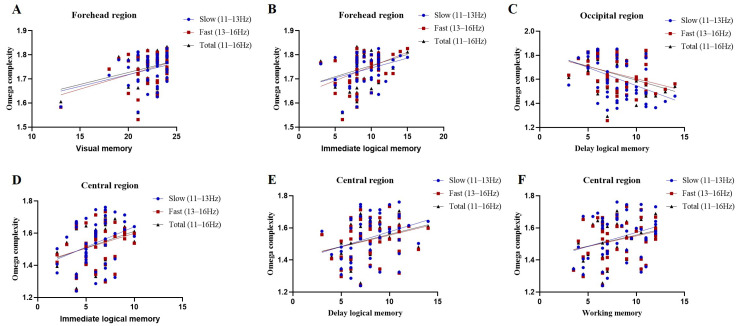
Correlation results of sleep spindle omega complexity and memory function; (**A**) the relationship Omega with visual memory; (**B**) the relationship Omega with immediate logical memory; (**C**) the relationship Omega with delayed logical memory; (**D**) the relationship Omega with immediate logical memory; (**E**) the relationship Omega with i delayed logical memory; (**F**) the relationship Omega with working memory.

**Table 1 jcm-12-00634-t001:** Demographic and clinical characteristics.

Parameters	Control Group (*n* = 59)	OSAS Group (*n* = 60)	t/χ^2^ Value	*p*-Value
Age (years)	36.42 (7.72)	38.28 (6.87)	−1.389	0.167
Education, years	16 (12,18)	16 (15,16)	−0.514	0608
BMI (Kg/m^2^)	23.89 (22.20, 25.50)	26.85 (25.15, 29.10)	−5.704	<0.001 *
ESS score	6.00 (4.00, 8.00)	7.00 (5.25, 10.75)	−3.151	0.002 *
Drink	10 (16.9)	8 (13.3)	0.303	0.582
Smoking	5 (8.5)	6 (10.0)	0.083	0.774
Hypertension	10 (16.9)	17 (28.3)	2.198	0.138
Diabetes	2 (3.4)	5 (8.3)	0.572	0.449
Apnea was witnessed	27 (45.8)	41 (68.3)	6.188	0.013 *
Suppress awake at night	20 (33.9)	24 (40.0)	0.475	0.491
Urinate night	19 (32.2)	21 (35.0)	0.104	0.747
Dream	33 (55.9)	33 (55.0)	0.010	0.919
Nightmare	5 (8.5)	2 (3.3)	0.643	0.422
Fatigue morning	24 (40.7)	33 (55.0)	2.445	0.118
Headaches morning	5 (8.5)	8 (13.3)	0.722	0.396
Dry mouth	24 (40.7)	40 (66.7)	8.083	0.004 *
Drowsiness	23 (39.0)	40 (66.7)	9.151	0.002 *
Memory deterioration	17 (51.7)	34 (56.7)	9.424	0.002 *
Early awakening	8 (13.6)	7 (11.7)	0.097	0.756

Note: BMI, Body Mass Index; ESS, Epworth Sleepiness Scale; *, *p* < 0.05.

**Table 2 jcm-12-00634-t002:** Memory function results for OSAS and control groups.

Parameters	Control Group (*n* = 59)	OSAS Group (*n* = 60)	t/Z Value	*p*-Value
DOT (mins)	9.00 (7.00, 10.50)	8.0 (6.50, 10.50)	−1.68	0.093
LM instant (mins)	5.0 (4.00, 6.00)	6.0 (5.00, 7.00)	−2.38	0.017 *
LM delay (mins)	4.00 (3.00, 5.00)	4.0 (3.00, 6.00)	−0.62	0.535
PRM Instant (s)	1406.92 (1265.38, 1593.50)	2002.67 (1826.75, 2280.67)	−7.58	<0.001 *
PRM delay (s)	1574.33 (1355.75, 1754.25)	1933.83 (1817.35, 2129.31)	−7.27	<0.001 *
SRM (s)	1437.20 (1175.60, 1744.00)	1663.50 (1310.95, 2020.95)	−2.26	0.023 *
SWM	27.00 (9.00, 31.00)	19.00 (9.25, 28.00)	−1.73	0.083

Note: *, *p* < 0.05; DOT, Digit Ordering Test; LM, Logical Memory; PRM, Pattern Recognition Memory; SRM, Spatial Recognition Memory; SWM, Spatial Working Memory.

**Table 3 jcm-12-00634-t003:** Power spectral density results for OSAS and control groups.

		F3	F4	C3	C4	O1	O2
11–13	Control	1.07 ± 0.38	1.11 ± 0.40	0.97 ± 0.75	0.92 ± 0.38	1.22 ± 0.46	1.15 ± 0.38
	OSAS	0.72 ± 0.43	0.75 ± 0.45	0.58 ± 0.35	0.61 ± 0.36	0.90 ± 0.88	0.78 ± 0.52
	t (*p*)	4.91 (0.000)	3.88 (0.000)	3.15 (0.002)	3.84 (0.000)	2.09 (0.04)	3.82 (0.000)
13–16	Control	0.83 ± 0.29	0.85 ± 0.32	0.73 ± 0.45	0.72 ± 0.28	0.93 ± 0.35	0.88 ± 0.25
	OSAS	0.61 ± 0.38	0.63 ± 0.38	0.50 ± 0.31	0.52 ± 0.32	0.76 ± 0.79	0.64 ± 0.43
	t (*p*)	3.04 (0.003)	2.91 (0.005)	2.87 (0.005)	3.10 (0.003)	1.29 (0.20)	3.14 (0.002)
11–16	Control	0.97 ± 0.33	1.00 ± 0.37	0.88 ± 0.60	0.84 ± 0.35	1.09 ± 0.37	1.03 ± 0.28
	OSAS	0.65 ± 0.39	0.68 ± 0.40	0.53 ± 0.32	0.56 ± 0.33	0.82 ± 0.82	0.70 ± 0.46
	t (*p*)	4.08 (0.000)	3.89 (0.000)	3.44 (0.001)	3.95 (0.000)	1.95 (0.06)	3.97 (0.000)

## Data Availability

The study data can be accessed from the corresponding author AC by request.

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
