# Peer review of "Sleep Spindle Characteristics and Relationship with Memory Ability in Patients with Obstructive Sleep Apnea-Hypopnea Syndrome"

_jcm, 2023, doi:10.3390/jcm12020634_

Round 1

Reviewer 1 Report

Zhu and colleagues in their work undertook an interesting topic of objective characterization with particular specific PSG characteristics and evaluation of the association with cognitive impairment in OSA patients, which is an increasing comorbidity present in this group of patients. Therefore, the search for new objective characteristics can have an effect on real-life practice. The results offer new insight into the topic.

The abbreviation used throughout the manuscript, OSAHS, should be expanded to Obstructive sleep apnea-hypopnea syndrome, or instead, the abbreviation OSAS should be used.

Please offer some explanation for the AHI<15 to be the cut-off point for the control group, add references for this as the interval 5-15 is already an OSA diagnosis, just the severity is mild. 

Why the inclusion criteria for age started from 23 y.o.?

Please add a reference for the criteria used for PSG scoring.

Was the normality of the data assessed? If so, please add the information to the statistical analysis section.

In table 1, please add years as a unit for age. 

In the description of all tables, add the expansion of all abbreviations used in each table. 

Changes in spectral density within age and gender are available in doi: 10.1111/jsr.12848 and could be discussed in the manuscript. 

Author Response

Dear  Reviewer

Thank you very much for your comments, please see the attachment

Reviewer 2 Report

The study by Zhu and colleagues investigated spindle characteristics and memory in patients with obstructive sleep apnea hypoventilation syndrome (OSAHS) and a control group. Participants completed 5 memory tasks in the evening, before sleeping overnight with polysomnography. Most notably, power spectral density in the fast and slow spindle bands was reduced in the patient group as compared to controls. Moreover, spindle density and complexity was linked to some measures of memory performance.

***

I found the topic of the manuscript to be interesting, and I commend the authors for conducting important research in this area. However, I have major concerns that must be addressed before this manuscript is suitable for publication. I have listed my comments below:

1.     It is unclear how the spindle power spectral density analysis was performed. Typically, spindle analyses are restricted to sleep stages N2 and/or N3. However, based on the Methods section it is my understanding that the authors did not score the sleep data before performing the spectral analysis. Accordingly, the analysis has been performed on the entire EEG recording (i.e., inclusive of N1, N2, N3, and REM sleep), rather than being restricted to the sleep stages of interest (N2 and N3 in this instance, given the prevalence of spindles in these stages). If the authors did in fact score the sleep data and restrict their analyses to sleep stages N2 and/or N3, this should be clarified in the manuscript. Otherwise, the analysis is suboptimal, and should be re-run whilst focusing on relevant sleep stages.

2.     Sleep spindles are widely believed to facilitate memory by driving memory consolidation (in concert with other oscillations that occur during non-REM sleep, namely slow oscillations and hippocampal ripples). Accordingly, studies investigating the role of sleep spindles in memory consolidation typically measure memory performance before and after an interval of sleep to ascertain the potential contribution of spindles to the change in memory performance. Critically, however, the authors here measured memory in a single session, which took place before the overnight sleep interval. From a theoretical perspective, it is unclear why the characteristics of spindles that occur after a learning/memory test would be associated with performance in the prior test?

3.     Throughout the Results section, the authors perform many analyses, but fail to adjust their alpha level to account for multiple comparisons. For example, when looking at the relationship between spindle complexity and memory ability, the authors look at three spindle types (fast, slow, total) across three brain regions (prefrontal, central, occipital) in relation to seven different measures of memory performance (see Table 2). Accordingly, for this particular analysis, the authors looked at 63 individual relationships. This is problematic, given that there is 5% probability of erroneously rejecting the null hypothesis in each individual test. The authors should correct for multiple comparisons throughout their Results section.

4.     In the Introduction (line 52), the authors cite work demonstrating that spindle density and frequency is reduced in OSAHS patients vs controls. I appreciate that the authors aimed to extend this existing literature by examining the oscillatory characteristics of sleep spindles in OSAHS. However, I believe it would be valuable to perform a spindle detection analysis (e.g. Tsanas & Clifford, 2015) to investigate whether spindle density/frequency was also lower in OSAHS patients in this dataset.

5.     It is unclear how the sample was selected. The authors report that data was collected from 145 patients who received PSG monitoring due to snoring (line 84). The manuscript reports data from 119 patients (line 89). The authors explain why data was excluded from 19 patients (line 88), but it’s unclear why an additional 7 patients were excluded. The authors should make this clear.

6.     Related to my previous point, eight female participants were excluded, but the authors provide no explanation for why they were excluded. Were the authors interested exclusively in male participants? If so, why?

7.     On line 122, the authors write that their study primarily used memory tests “related to memory consolidation”. However, performance in the memory tests used here is affected by encoding, consolidation, and retrieval processes (i.e., none of them isolate the consolidation process). Accordingly, performance does not relate directly to memory consolidation per se.

8.     In Table 2, the authors report the results of their memory function tests. For their measure of immediate logical memory (LM Instant (mins)), the mean values reported are identical (5.00) in both groups, yet the p-value is reported as 0.017. I assume this is a mistake?

9.     Figure 2 is erroneously labelled as Figure 3.

Author Response

(The authors gave the same response as above.)
